# Sample what you can't compress

## Abstract

For learned image representations, basic autoencoders often produce blurry results. Reconstruction quality can be improved by incorporating additional penalties such as adversarial (GAN) and perceptual losses. Arguably, these approaches lack a principled interpretation. Concurrently, in generative settings diffusion has demonstrated a remarkable ability to create crisp, high quality results and has solid theoretical underpinnings (from variational inference to direct study as the Fisher Divergence). Our work combines autoencoder representation learning with diffusion and is, to our knowledge, the first to demonstrate *jointly learning a continuous encoder and decoder under a diffusion-based loss and showing that it can lead to higher compression and better generation.*. We demonstrate that this approach yields better reconstruction quality as compared to GAN-based autoencoders while being easier to tune. We also show that the resulting representation is easier to model with a latent diffusion model as compared to the representation obtained from a state-of-the-art GAN-based loss. Since our decoder is stochastic, it can generate details not encoded in the otherwise deterministic latent representation; we therefore name our approach "Sample what you can't compress", or SWYCC for short.

## 1 Introduction

Image autoencoders ultimately necessitate a pixel-level loss to measure and minimize distortion. A common choice is to use mean squared error (MSE). This is a problem for image and video models because MSE favors low frequencies over high frequencies (Rybkin, 2018). Although generalized robust loss functions have been developed (Barron, 2019), they are insufficient on their own for avoiding blurry reconstructions. A popular fix is to augment a pixel-level loss with additional penalties. Typically, MSE is still used because it is easy to optimize due to its linear gradient.

For example, Rombach et al. (2021) use a combination of MSE, perceptual loss, and adversarial loss. Esser et al. (2021) noted that an adversarial loss helps them get high-quality images with realistic textures. Unfortunately GANs remain challenging to train; which was most recently noted by Kang et al. (2023), when they couldn't naively scale up their architecture. The diversity of their outputs is also limited, because modern GAN based decoders are deterministic, and thus lack the capacity to sample multiple different possibilities.

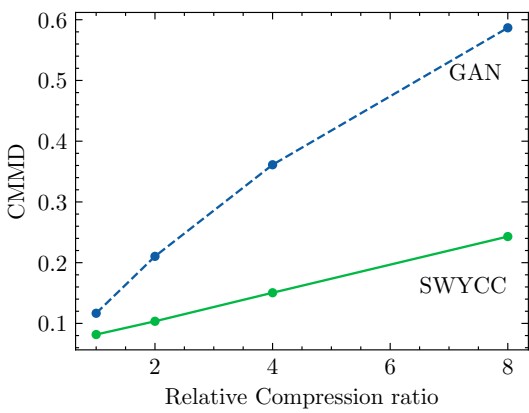

Figure 1: Reconstruction distortion (lower is better) as a function of compression for SWYCC and GAN based auto-encoders.

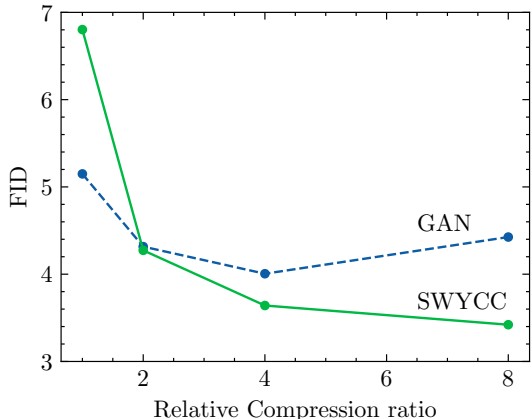

Figure 2: Class conditional generation quality (lower is better) as a function of compression for SWYCC and GAN based auto-encoders.

As an alternative, this paper describes a technique for using a diffusion loss to learn an autoencoder. The diffusion loss is sensible because it is a proper scoring rule with favorable theoretical properties such as being formally connected to KL divergence (Sohl-Dickstein et al., 2015). It has proven itself capable of generating crisp results with high perceptual quality as reflected by human evaluation studies (Hoogeboom et al., 2023b; Karras et al., 2024b).

To demonstrate its simplicity, we take a popular encoder architecture Chang et al. (2022) and marry it with a U-Net decoder, a popular denoising architecture for diffusion (Hoogeboom et al., 2023b). With some additional details outlined in section 2, we show that this approach achieves lower distortion at all compression levels as measured by the CMMD metric (Jayasumana et al., 2024). Because our decoder is able to sample details at test-time that are not encoded in the latents, we call our approach "Sample what you can't compress" or SWYCC for short. This work will show that,

- SWYCC achieves lower reconstruction distortion at all tested compression levels vs SOTA GAN-based autoencoders (section 3).
- SWYCC representations enable qualitatively better latent diffusion generation results at higher compression levels vs SOTA GAN-based autoencoders (section 3.3).
- Splitting the decoder into two parts improves training dynamics (section 3.1 and 3.2).

## 2 METHOD

Eliding its various parametrizations for brevity, the standard diffusion loss is characterized by the Monte Carlo approximation of the following loss,

$$\ell(x) \stackrel{\text{def}}{=} \mathsf{E}_{\varepsilon \sim \text{MVN}(0, I_{h \cdot w \cdot 3}), t \sim \text{Uniform}[0,1]} \left[ w_t \left\| x - D(\alpha_t x + \sigma_t \varepsilon, t) \right\|_2^2 \right]. \tag{1}$$

Herein $x \in \mathbb{R}^{h \times w \times 3}$ denotes a natural image and $D$ is a neural network fit using gradient descent and which serves to denoise the corrupted input $x_t \stackrel{\text{def}}{=} \alpha_t x + \sigma_t \varepsilon$ at a given noise-level $t$. Let the corruption process be the cosine schedule (Hoogeboom et al., 2023a), $\sigma_t^2 \stackrel{\text{def}}{=} 1 - \alpha_t^2$ and $\alpha_t \stackrel{\text{def}}{=} \cos(at + b(1-t))$ where $a = \arctan(e^{10})$ and $b = \arctan(e^{-10})$.

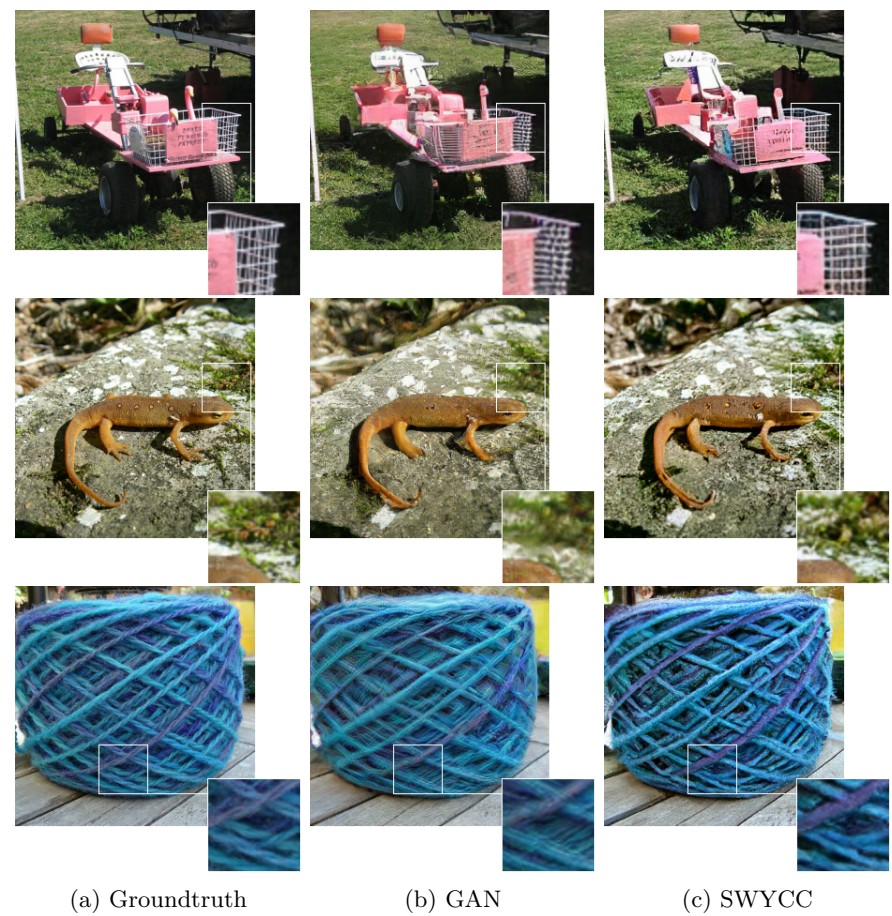

(a) Groundtruth          (b) GAN          (c) SWYCC

Figure 3: Comparison of GAN versus SWYCC reconstructions at $8\times$ relative compression level. You can see that the GAN based autoencoder loses a significant amount of detail in the highlighted portions, which SWYCC is able to sample effectively.

We extend this definition to the task of autoencoding by simply allowing the denoising function to take an additional argument, $D_{\text{Initial}}(E(x))$, itself having access to the uncorrupted input $x$ but only through the bottlenecking function $E$. The result is,

$$\ell_{\text{AE}}(x) \overset{\text{def}}{=} \mathsf{E}_{\varepsilon \sim \text{MVN}(0, I_{h \cdot w \cdot 3}), t \sim \text{Uniform}[0,1]} \left[ w_t \left\| x - D_{\text{Refine}}(\alpha_t x + \sigma_t \varepsilon, t, D_{\text{Initial}}(E(x))) \right\|_2^2 \right]. \quad (2)$$

As the notation suggests $E$ is an encoder which, notably, is learned jointly with "diffusion decoder" $D_{\text{Refine}}$ and secondary decoder $D_{\text{Initial}} : \mathcal{Z} \to \mathbb{R}^{h \times w \times 3}$. The specification of $D_{\text{Initial}}$ is largely a convenience but also merits secondary advantages. By mapping $z = E(x)$ back into $x$-space, we can simply concatenate the corrupted input $x_t$ and its "pseudo reconstruction," $D_{\text{Initial}}(z)$. Additionally, we find that directly penalizing $D_{\text{Initial}}(z)$, as described below, speeds up training.

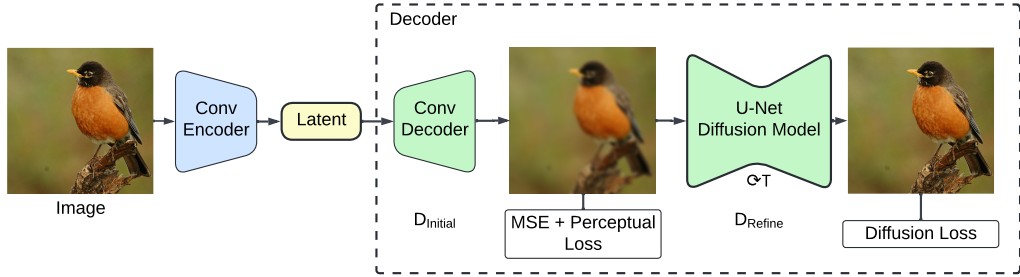

Figure 4: A block diagram of our auto-encoder architecture. Our "Diffusion Model" is a U-Net as defined by Hoogeboom et al. (2023b). During inference, the diffusion model is run in a loop to iteratively sample an image. All of these parameters jointly after being initialized from scratch, except for the weights of the network used in the perceptual loss.

## 2.1 ARCHITECTURAL DETAILS

**Encoder:** We use a fully convolutional encoder in all of our experiments, whose specifics we borrow from MaskGIT(Chang et al., 2022). The encoder consists of multiple ResNet(He et al., 2016) blocks stacked on top of each other, with GeLU(Hendrycks & Gimpel, 2017) for its non-linearities and GroupNorm(Wu & He, 2018) for training stability. The ResNet blocks are interspersed with strided convolutions with stride 2 which achieves a $2\times$ downsampling by itself. To get the $8 \times 8$ patch size, we use 4 ResNet blocks with 3 downsampling blocks. The encoder architecture is common for all of our experiments, and we only change the number of channels at the output layer to achieve the desired compression ratio.

**Decoder:** For the decoder in the GAN baseline and $D_{\text{Initial}}$ we use an architecture that is the reverse of the encoder. For up-sampling, we use the depth-to-space operation. Just like the encoder, we have 4 ResNet blocks interspersed with 3 depth-to-space operations. For $D_{\text{Refine}}$ we use a U-Net as defined by Hoogeboom et al. (2023b). The U-Net has 4 ResNet blocks for downsampling and corresponding 4 ResNet blocks for upsampling with residual connections between blocks of the same resolution. After 4 downsampling stages, when resolution is $16 \times 16$, we use a self-attention block to give the network additional capacity.

## 2.2 REDUCING DISTORTION USING ADDITIONAL DISTANCE METRICS

We find that additional direct penalization of $D_{\text{Initial}}(E(x))$ leads to improved CMMD and FID and less distortion (see Figure 11). This was achieved by minimizing a composite loss containing terms with favorable Hessian (eq. (4)) and perceptual characteristics (eq. (5)),

$$\ell_{\text{Total}} \overset{\text{def}}{=} \ell_{\text{AE}} + \lambda_p \ell_{\text{Perceptual}} + \lambda_m \ell_{\text{MSE}} \tag{3}$$

where,

$$\ell_{\text{MSE}} \overset{\text{def}}{=} \|x - D_{\text{Initial}}(E(x))\|_2^2 \tag{4}$$

and,

$$\ell_{\text{Perceptual}} \overset{\text{def}}{=} \left\| f_{\text{Frozen}}(x) - f_{\text{Frozen}}\left(D_{\text{Initial}}(E(x))\right) \right\|_2^2. \tag{5}$$

The function $f_{\text{Frozen}}$ is an unlearnable standard ResNet, itself trained on ImageNet and used for both the baseline and SWYCC. We found the best hyper-parameter setting for eq. (3) is $\lambda_m = 1$

| $\lambda_p$ | $\lambda_m$ | CMMD ↓ |
|---|---|---|
| 0 | 0 | 0.43 |
| 0 | 1 | 0.32 |
| 0.1 | 0 | 0.13 |
| 0.1 | 1 | 0.15 |

Table 1: Individual impact of each of our auxiliary losses. We note that the perceptual loss has a big impact, and is crucial to being competitive while training autoencoders. See image samples in Figure 11. The first row $\lambda_p = \lambda_m = 0$ is analogous to $D_{\text{Initial}}$ having no role to play.

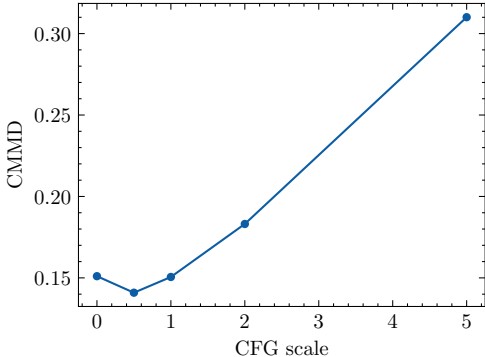

Figure 5: Effect of CFG scale in $D_{\text{Refine}}$. We use CFG scale of 0.5 in our experiments.

and $\lambda_p = 0.1$ Setting $\lambda_p > 0$ was particularly important to be competitive at reconstruction with GAN based methods in Table 1. The visual impact of perceptual loss is shown in Figure 11.

For generating reconstructions (recall that the SWYCC decoder is stochastic) we used classifier-free guidance during inference (Ho & Salimans, 2021); for the unconditional model the U-Net was trained with $D_{\text{Initial}}(E(x))$ dropped-out, i.e., randomly zeroed out on 10% of training instances.

## 3 EXPERIMENTS

In this section we explore how the GAN-based loss compares to our approach. Without loss of generality, we define the relative compression ratio of 1 to be a network that maps $8 \times 8$ RGB patches to an 8 dimensional latent vector. Effectively, this means for our encoder $E$ if and $x \in \mathbb{R}^{256 \times 256 \times 3}$, then $E(x) \in \mathbb{R}^{32 \times 32 \times C}$ where $C = 8$. In general, to achieve a relative compression ratio of $K$ we set $C = \frac{8}{K}$. The effect of increasing the compression ratio is plotted in Figure 1. Observe that distortion degrades much more rapidly for the GAN based auto-encoder as measured by CMMD (Jayasumana et al., 2024) which Imagen-3 (Imagen-Team-Google et al., 2024) showed better correlates with human perception.

Not only is our approach better at all compression levels; the gap between the GAN based autoencoder and SWYCC widens as we increase the relative compression ratio. Using the much simpler diffusion formulation under Equation 3, we are able to reconstruct crisp looking images with detailed textures (See Figure 3). Our method has the added benefit that we do not need to tune any GAN related hyper-parameters, and can scale up effectively using the large body of diffusion literature (Karras et al. (2022), Karras et al. (2024a)).

### 3.1 IMPACT OF $D_{\text{Initial}}$

Observing Equation 2 and Figure 4, we note that the output of $D_{\text{Initial}}$ is an intermediate tensor that is not strictly required for the diffusion loss or for generating the output. We show using Table 1 that this piece is crucial for achieving performance comparable to the GAN based autoencoder. The perceptual loss term in particular has a large impact in reducing distortion. Visual examples are shown in Figure 11. We found that separating the decoder into 2 parts was necessary. When we

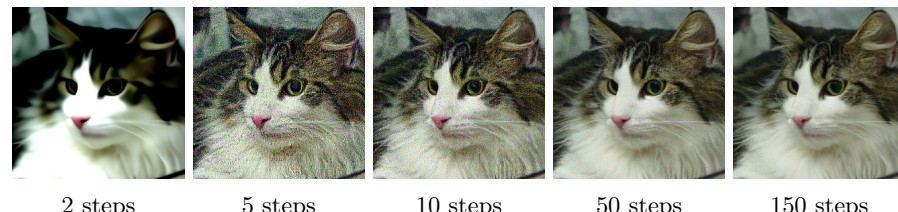

| 2 steps | 5 steps | 10 steps | 50 steps | 150 steps |

Figure 6: Example of how reconstructions look when changing the number of sampling steps for $D_{\text{Refine}}$. We can see that even with steps $\leq 5$, the high level structure of the image is preserved. When our method is used for interactive generation (for example, with a latent diffusion model), we can use fewer steps of $D_{\text{Refine}}$ to show the user multiple low-quality inputs, and use a high number of steps for the final generation that the user selects.

applied LPIPS and MSE losses without $D_{\text{Initial}}$, our method was not competitive with GAN based autoencoders.

## 3.2 Analysis of sampling in $D_{\text{Refine}}$

In Figure 6 we show the qualitative difference number of sampling steps makes to reconstructions. We can see that even with just 2 steps the high level structure of the image is present. In Figure 7 we study the impact of number of sampling steps by using the CMMD metric. Figure 9 shows what sampling in $D_{\text{Refine}}$ actually ends up changing. We can see that only regions with high-frequency components and detailed textures are changed between samples, while regions containing similar colors over large areas are left untouched.

Figure 5 studies the effect of classifier-free guidance (Ho & Salimans, 2021) as used in $D_{\text{Refine}}$. We ablate the guidance with a model trained at a relative compression factor of 4 (See Section 3 for definition) and find that a guidance scale of 0.5 works the best. This is not to be confused with the guidance scale of the latent diffusion model that may be trained on top of our autoencoder, which is a completely separate parameter to be tuned independently.

## 3.3 Modeling latents for diffusion

We use a DiT model (Peebles & Xie, 2023) to model the latent space of our models for the task of class-conditional image generation. In Figure 2 we compare our latents with those of a GAN based autoencoder. Our approach leads to 5% lower FID (Heusel et al., 2017) than the GAN baseline at the best $4\times$ compression ratio. Notably our approach achieves its best result at the highest compression ratio, where the task of modelling the latent representation is simplest, whereas the GAN autoencoder is unable to operate effectively in this regime.

## 3.4 Exploring better perceptual losses

In Table 1 we showed the large impact perceptual loss has on reconstruction quality. This begs the question; are there better auxiliary losses we can use? We compare the perceptual loss as described by in VQGAN (Esser et al., 2021; Johnson et al., 2016) and replace it with the DISTS (Ding et al., 2020). DISTS loss differs from perceptual loss in 2 important ways. a) It uses SSIM (Wang et al., 2004) instead of mean squared error and b) It uses features are multiple levels instead of using only the activation's from the last layer. The results are shown in Figure 10. At lower relative

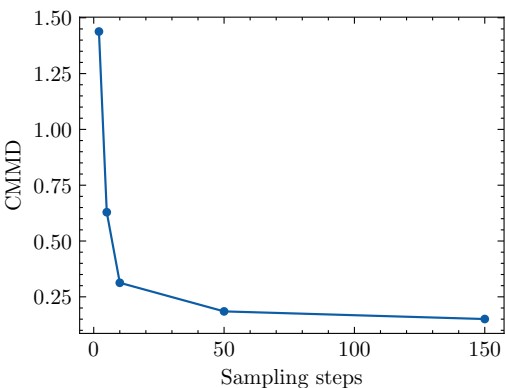

Figure 7: Impact of number of sampling steps on $D_{\text{Refine}}$ on CMMD.

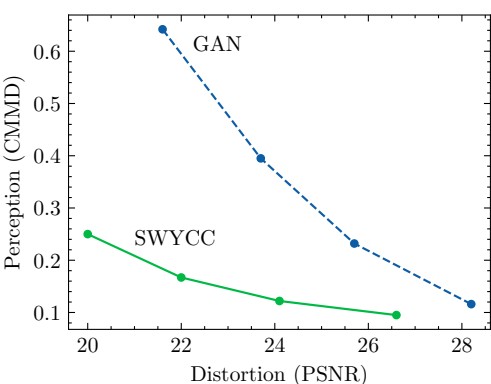

Figure 8: Rate-distortion-perception trade-off as outlined by Blau & Michaeli (2019). Despite higher distortion, SWYCC is perceptually better.

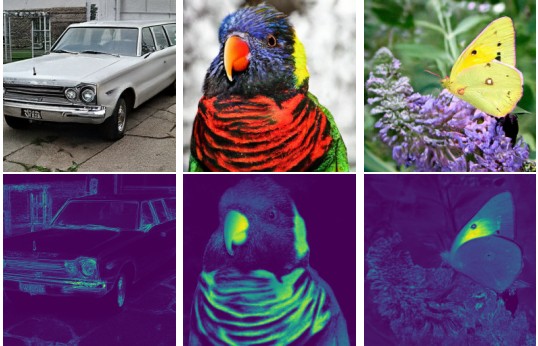

Figure 9: Reconstructed images and a heat-map of variance between 10 samples.

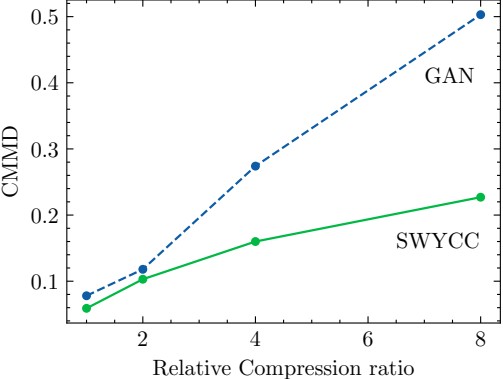

Figure 10: Effect of using DISTS loss on reconstruction quality.

compression ratios, DISTS loss helps GANs and SWYCC. But at higher relative compression ratios, GAN based autoencoders perform even worse than the perceptual loss based baseline. We think this is an avenue for future exploration. We perform all other experiments with perceptual loss (Esser et al., 2021), since it is more prevalent in literature and it helps GAN based auto-encoders at higher relative compression ratios.

### 3.5 ARCHITECTURE AND HYPER-PARAMETERS

**Autoencoder training hyper-parameters** We train all of our models on the ImageNet dataset resized at $256 \times 256$ resolution. During training, we resize the image such that the shorter side measures 256 pixels and take a random crop in that image of size $256 \times 256$. For measuring reference statistics on the validation set, we take the largest possible center square crop. All of our models are trained at a batch size of 256 for $10^6$ steps which roughly equals 200 epochs.

**GAN baseline:** We use the popular convolutional encoder-decoder architecture popularized by MaskGIT (Chang et al., 2022) in our GAN-based baselines. This autoencoder design is used by many image models, including FSQ (Mentzer et al., 2024) and GIVT (Tschannen et al., 2023), and was extended to the video domain by MAGVIT-v2 (Yu et al., 2024), VideoPoet (Kondratyuk et al., 2024), and WALT (Gupta et al., 2023). We take advantage of decoder improvements developed by MAGVIT-v2 (see Section 3.2 in (Yu et al., 2024)) to improve reconstruction quality. Note that while the video models enhance the base autoencoder architecture with 3D convolution to integrate information across time, the discriminator and perceptual loss are still applied on a per-frame basis and thus are essentially unchanged in our model.

**SWYCC:** In our experiments we keep the architecture of $D_{\text{Initial}}$ identical to the decoder used in the GAN baseline. For $D_{\text{Refine}}$ we use the U-Net architecture as parameterized by Hoogeboom et al. (2023b). We borrow the U-Net 256 architecture and make the following modifications:

```
channel_multiplier = [1, 2, 4, 8]
num_res_blocks = [2, 4, 8, 8]
downsampling_factor = [1, 2, 2, 2]
attn_resolutions = [16]
dropout = 0.0
```

We train using $v$-parameterization (Salimans & Ho, 2022), which corresponds to $w_t = \sigma_t^{-2}$ in equation 1. We use the Adam optimizer to learn our parameters. The learning rate is warmed up for $10^4$ steps from 0 to a maximum value of $10^{-4}$ and cosine decayed to 0. We use gradient clipping with global norm set to 1.

**Latent Diffusion**: All of our experiments are done with the DiT-L architecture with $2 \times 2$ patching (Peebles & Xie, 2023) with the addition of SwiGLU (Shazeer, 2020) and 2D RoPE (Heo et al., 2024). We train for $4 \times 10^5$ steps with a batch size of 256, using a constant learning rate of $10^{-4}$ and dropout the class embedding 10% of the time during training. For inference we use a classifier-free guidance scale of 0.5 which gave optimal results in Peebles & Xie (2023).

## 4 RELATED WORK

**Autoencoders for 2-stage generation:** For discrete representation learning, van den Oord et al. (2017) showed the usefulness of the 2-stage modeling approach. In this broad category, the first stage fits an autoencoder to the training data with the goal of learning a compressed representation useful for reconstructing images. This is followed by a second stage where the encoder is frozen and a generative model is trained to predict the latent representation based on a conditioning signal. This approach regained popularity when Ramesh et al. (2021) showed that it can be used for zero-shot text generation, and is now the dominant approach for image and video generation (Chang et al., 2022; Yu et al., 2024; Chang et al., 2023; Gupta et al., 2023; Kondratyuk et al., 2024).

**Adversarial losses:** Esser et al. (2021) extended the autoencoder from van den Oord et al. (2017) with two important new losses, the perceptual loss and the adversarial loss, taking inspiration from the works of Johnson et al. (2016) and Isola et al. (2017). The perceptual loss is usually defined as the L2 loss between a latent representation of the original and reconstructed image. The latent representation, for example, can be extract from the final layer activations of a ResNet optimized to classify ImageNet images. The adversarial loss is a patch-based discriminator that uses a discriminator network to predict at a patch-level whether it is real or fake. This encourages the decoder to produce realistic looking textures.

**Latent Diffusion:** Rombach et al. (2021) popularized text-to-image generation using latent diffusion models. They kept the autoencoder from Esser et al. (2021) intact and simply removed the quantization layer. This accelerated diffusion model research in the community owing to the fact that the latent space was much smaller than the pixel space, which allows fast training and inference compared to diffusion models like Imagen that sample pixels directly (Saharia et al., 2022).

**2-stage diffusion autoencoders:** Pandey et al. (2022) and Preechakul et al. (2022) both train decoders with a diffusion loss. The crucial difference is that both these works train their autoencoder in 2-stages.

**Compression and diffusion:** Hoogeboom et al. (2023a) showed that diffusion models can be used for compression. Crucially, compared to our approach they use a frozen autoencoder, and do not train their autoencoder end-to-end. They also use an objective based on modified flow matching. In contrast, we did not modify the loss or the sampling algorithm. Pandey et al. (2022) use a similar approach with a 2-stage autoencoder training process for their autoencoder.

In a similar context, Yang & Mandt (2024) developed an end-to-end optimized compression model using a diffusion decoder. They show improved perceptual quality compared to earlier GAN-based compression methods at the expense of higher distortion (pixel-level reconstruction accuracy). Different from our approach, they use a discrete latent space, which is required for state-of-the-art compression rates achieved via entropy coding. This limits the reconstruction quality but is required for a compression model that ultimately seeks to minimize a rate-distortion objective, not just a reconstruction and sampling quality objective.

Würstchen architecture (Pernias et al., 2024) has shown that training a cascade of diffusion models improves training efficiency. Crucially Würstchen, does not apply a diffusion loss on pixels, instead resorting to a GAN loss. Whang et al. (2022) also pointed out that diffusion can fix a lot of pixel level artifacts, although they do not investigate training autoencoders.

Shi et al. (2022) learn an autoencoder using a diffusion loss for discrete vector quantized encodings. We differ from them in 2 crucial ways; (i) we learn a continuous representation (ii) We show that our architecture produces latents that are better for latent diffsion by showing that we can achieve *higher compression and better generation performance (Figure 2)* .

## 5 Conclusion

We have described a general autoencoder framework that uses a diffusion based decoder. Compared to decoders that use GANs, our system is much more easier to tune and has the same theoretical underpinnings as diffusion models. We showed our method produces sigificantly less distortions as compared to GAN based autoencoders in Figure 1 and are better behaved as latent spaces for diffusion in Figure 2. In Section 3.1 and 3.2 we studied the hyper-parameter settings on the 2 major components of our decoder, $D_{\text{Initial}}$ and $D_{\text{Refine}}$.

**Possible extensions:** The autoencoder technique we describe is fairly general and can be extended to any other continuous modality like audio, video or 3D-point clouds. In addition, all improvements to diffusion algorithms like those by Karras et al. (2024a) can be carried over.

**Limitations:** The main limitation of our method is the increase in inference cost during decoding. This can be partly mitigated by using fewer steps like in Figure 6. In addition, techniques used to improve diffusion sampling time like Progressive distillation (Salimans & Ho, 2022) and Instaflow (Liu et al., 2024) are also prudent. Because of $D_{\text{Refine}}$, our training time compute cost is also higher. Combining $D_{\text{Initial}}$ and $D_{\text{Refine}}$ in a clever way to reduce training time compute and memory could be a promising research direction.

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

# 6 APPENDIX

## 6.1 ADDITIONAL METRICS

| Method | Rel. Compression | rFID | Inception Score | pSNR | CMMD | rFID(Dino-v2) |
|---|---|---|---|---|---|---|
| SWYCC | 1 | 0.33 | 222 | 26.6 | 0.095 | 4.13 |
| | 2 | 0.62 | 215 | 24.1 | 0.122 | 9.56 |
| | 4 | 1.17 | 203 | 22.0 | 0.167 | 22.4 |
| | 8 | 2.75 | 175 | 20.0 | 0.250 | 60.1 |
| GAN | 1 | 0.27 | 222 | 28.2 | 0.116 | 4.52 |
| | 2 | 0.54 | 215 | 25.7 | 0.232 | 11.7 |
| | 4 | 0.99 | 202 | 23.7 | 0.395 | 30.3 |
| | 8 | 1.96 | 176 | 21.6 | 0.642 | 72.9 |
| SD-VAE 2.x (on COCO) | 2 | 4.70 | | 24.5 | | |

Table 2: Additional reconstruction metrics on ImageNet, unless otherwise noted. SWYCC is better than GAN at all perceptual metrics except rFID.

## 6.2 PARAMETER COUNTS

| Method | Component | Parameters (Million) |
|---|---|---|
| SWYCC | Encoder | 49.4 |
| | $D_{\text{Initial}}$ | 63.4 |
| | $D_{\text{Refine}}$ | 614.1 |
| GAN | Encoder | 49.4 |
| | Decoder | 63.4 |

Table 3: Parameter counts of network components.

705
706
707
708
709
710
711
712
713
714
715
716
717
718
719
720
721
722
723
724
725
726
727
728
729
730
731
732
733
734
735
736
737
738
739
740
741
742
743
744
745
746
747
748
749
750
751

### 6.3 Visual examples

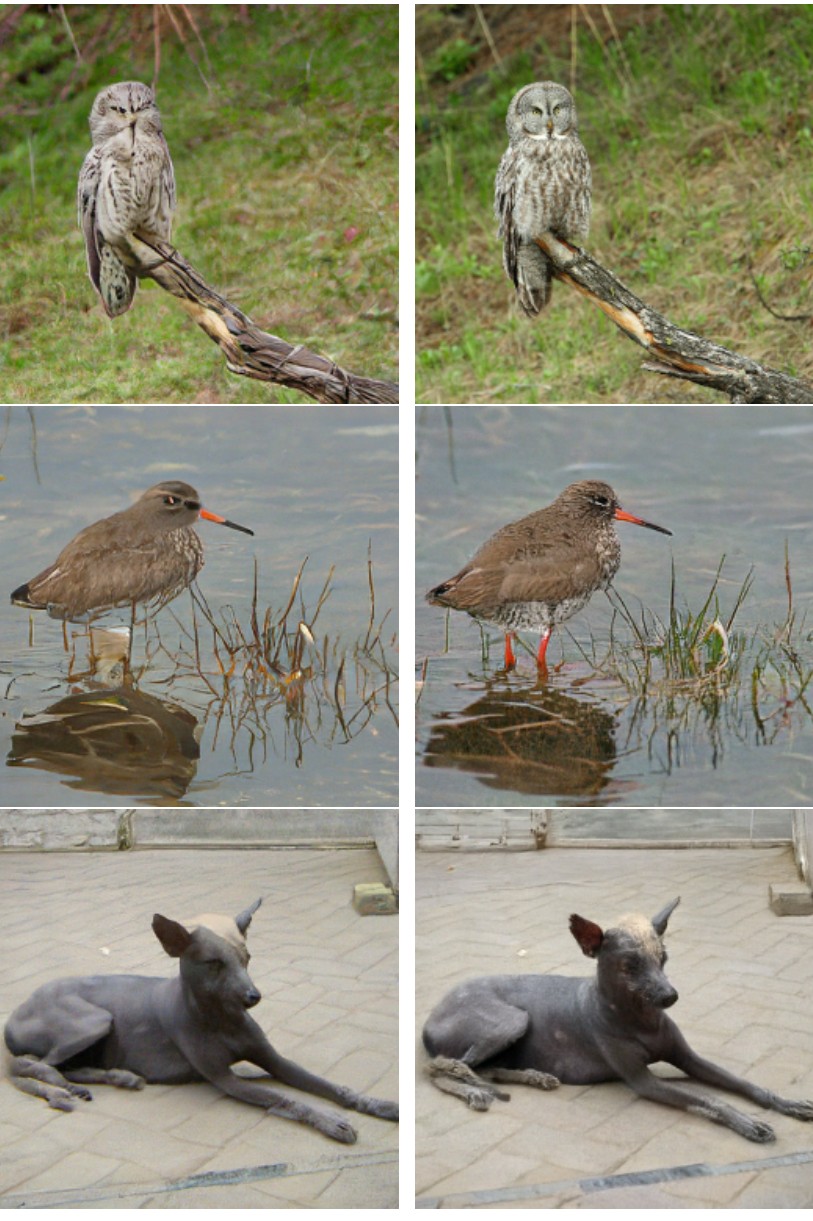

(a) No auxiliary losses.     (b) Only perceptual auxiliary loss.

Figure 11: Visual impact of perceptual loss.