# OpenReview forum: "Sample what you can't compress"
_ICLR.cc/2025/Conference — Submitted to ICLR 2025_

### Official Review · Reviewer_f11k · 2024-10-21

**Soundness:** 2
**Presentation:** 2
**Contribution:** 2
**Rating:** 3
**Confidence:** 5

**Summary:**

This paper proposes to use a diffusion model for image compression. Specifically, the diffusion model is an encoder-decoder architecture, where the encoder encodes the given clean image, and the decoder generates a high-quality image from pure noise (as in standard diffusion methods) while being conditioned on the encoded image. The encoder and the decoder are trained jointly using the standard MSE diffusion loss, together with some tweaks such as adding a perceptual loss after an intermediate decoding step.

**Strengths:**

1. The proposed approach for image compression is simple and effective. It makes sense that it should work better than GAN based methods, as diffusion models beat GANs in many different applications. GANs are indeed incredibly difficult to train.
2. The paper is overall clear and written well.
3. There are several ablation studies.

**Weaknesses:**

1. As far as I understand, the proposed approach is not novel. See [1] for example. The only differences that I see are some optimization/loss tweaks. If the authors could clarify the differences and why their approach is novel, that would be great. Currently, [1] is not discussed in the manuscript.

2. A comparison with several previous methods is missing, specifically a comparison on the rate-perception-distortion plane [2]. When designing compression methods that only aim for minimal distortion (e.g., MSE), we usually compare them on the rate-distortion plane, as the authors did in figure 8. However, when designing high-perceptual-quality compression methods, we usually compare them on the rate-perception-distortion plane, as these three desired forces are at odds with each other [2].

3. The authors only demonstrate their method on ImageNet. But what about simpler data sets, such as CelebA,CIFAR, etc.? Is the proposed approach still more effective than GANs?

4. The limitations section is very limited. The authors discuss only the limitation of almost all diffusion methods: requiring a large number of inference steps to produce high-quality images. What about model size, training time, required data set size, etc., as compared to GAN based methods?

[1] Konpat Preechakul et al., "Diffusion Autoencoders: Toward a Meaningful and Decodable Representation", CVPR 2022.

[2] Yochai Blau and Tomer Michaeli, "Rethinking Lossy Compression: The Rate-Distortion-Perception Tradeoff", ICML 2021

**Questions:**

1. Can the authors please explain the differences between the proposed approach and [1]?
2. Can the authors explain why there is no comparison with additional methods, and in particular why there is no comparison on the rate-perception-distortion plane [2]?


[1] Konpat Preechakul et al., "Diffusion Autoencoders: Toward a Meaningful and Decodable Representation", CVPR 2022.

[2] Yochai Blau and Tomer Michaeli, "Rethinking Lossy Compression: The Rate-Distortion-Perception Tradeoff", ICML 2021

---

> ### Author Response · Authors · 2024-11-15
> **DiffusionAE, metrics, datasets**
>
> (1) DiffusionAE:
> These is indeed relevant work, thanks for pointing these out.
> DiffusionAE section 4:
> “We first train the semantic encoder (φ) and the image decoder (θ) via Equation 6 until convergence. Then, we train the latent DDIM (ω) via Equation 9 with the semantic encoder fixed.”
>
> (2) Rate-perception-distortion
> It is indeed true that perceptual metrics and distortion metrics are at odd. We will include this comparison in the camera ready version.
>
> (3) Additional datasets:
> We are working  to see if we can get additional datasets evaluated.  Limitations:
> We elaborated more on training time compute in limitations. We will include a training time and model parameters table shortly.

---

> ### Comment · Reviewer_f11k · 2024-11-19
>
> I'd like to thank the authors for considering my review.
>
> (1) OK.
>
> (2) Without seeing a proper comparison on the rate-distortion-perception plane, unfortunately I can't know whether the proposed method is better or worse than previous ones.
>
> (3) OK. I am keen to see how does the proposed method performs on simpler data sets, where its tested in more "trivial" settings.

---

> > ### Author Response · Authors · 2024-11-23
> > **Additional metrics added, rate-distortion-perceptual tradeoff curve added**
> >
> > We added additional metrics in appendix and modified Figure 8 to show the rate-distortion-perceptual tradeoff. Please let us know if this is what you had in mind.
> >
> > Our hope is that you can reconsider your rating based on this.

---

### Official Review · Reviewer_97eW · 2024-10-27

**Soundness:** 3
**Presentation:** 2
**Contribution:** 1
**Rating:** 3
**Confidence:** 4

**Summary:**

This paper presents SWYCC, a VAE trained with diffusion loss. It optimizes performance by adjusting loss weights and sampling steps. The authors then compare it to VQGAN-based methods, demonstrating improved image reconstruction.

**Strengths:**

- The paper provides extensive detail on the architectural design of SWYCC, including the detail structure of the encoder and decoder. However, while this information is valuable for understanding the framework, it is not the primary contribution of the paper and does not introduce novel architectural concepts.

- The inclusion of a classifier-free guidance experiment is a notable strength, as this approach introduces a new aspect to the diffusion-based decoder. However, it is important to note that the classifier-free technique employed significantly increases computation time during the sampling stage, effectively doubling it, which also poses challenges in practical applications.

**Weaknesses:**

- The paper includes too few image comparisons. It does not show many comparisons between different model compressions, making it difficult to evaluate how SWYCC stands relative to other approaches.

- There is no comparison with Stable Diffusion XL's VAE, which is one of the widely used VAE models.

- The method relies solely on matrices for image generation, which is not common in image reconstruction compression. While it is acceptable for the authors to report measurements they find appropriate, it is also important to include other widely used metrics in the field, such as PSNR, LPIPS, and rFID.

- The paper lacks mention and discussion of previous diffusion-based autoencoders, such as DiffusionAE ([link](https://diff-ae.github.io/)) and DiVAE ([link](https://arxiv.org/abs/2206.00386)). Care should be taken not to claim to be the first when there are multiple works prior to this.

- Training the decoder under diffusion loss is a concept introduced in 2022. Although the idea of using diffusion as a decoder has been recognized for its benefits, it has not gained popularity compared to GAN-based training primarily due to its higher computational cost. This concern is not adequately addressed in the paper.

**Questions:**

- What does "favorable Hessian" mean? It doesn't seem to relate to MSE.
- How does this approach differ from fine-tuning a standard VAE and using diffusion for upscaling?
- How does "speed up training" be versify?

---

> ### Author Response · Authors · 2024-11-15
>
> (1) Image comparisons and SDXL:
> We are including a comparison with SDXL in the next revision.
>
> (2) Metrics:
> This is a vlid point and we are including a table with PSNR, LPIPS and rFID in the next revision.
>
> (3) DiffusionAE:
> These is indeed relevant work, thanks for pointing these out.
> DiffusionAE section 4:
> “We first train the semantic encoder (φ) and the image decoder (θ) via Equation 6 until convergence. Then, we train the latent DDIM (ω) via Equation 9 with the semantic encoder fixed.”
>
> (4) DiVAE:
> They do not study a continuous latent space, and moreover don’t show how their setup can achieve higher compression and better generation performance than the GAN based loss. We have modified the abstract to now say
> “to our knowledge, the first to demonstrate  jointly learning a continuous encoder and decoder under a diffusion-based loss
> and showing that it can lead to higher compression and better generation.”
>
> Note that in DiVAE Table 4 just like the GAN loss, their generative performance suffers with higher compression.
> If you don’t think that is justified, we can re-word it further.
> We have also elaborated on this in related works. I also could not find a conference or journal reference for them; please let us know if there is a better citation for them than arxiv.
>
> (5) Compute concerns:
> We have elaborated on this in the limitations section
>
> (6) Hessian:
> We meant to say that intuitively, if there is just MSE loss things are more stable. Mathematically, with an MSE loss, the condition number is bounded. This practically means that a single LR will work well in all epochs of training. If you think this deserves to go in the paper, we will elaborate there.
>
> (7) Difference from fine-tuning VAE:
> The main difference is learnings the encoder jointly. This is crucial for the main claim that we can achieve higher compression and better generation results.
>
> (8) Speed up training:
> This is a good point, we reworded that section to “reducing distortion using additional distance metrics”:

---

> ### Author Response · Authors · 2024-11-23
> **Additional comparisons added in Appendix**
>
> Please see Table 2

---

> ### Author Response · Authors · 2024-11-23
> **About comparison to SDXL**
>
> Note that SDXL's VAE is just captured in the GAN baseline @ 2x relative compression ratio (since they have 4 channels).
>
> We tuned the GAN baseline's parameters. Given that we are able to show that SWYCC is better than the GAN baseline, it follows by extension that our method is better than SDXL's VAE.

---

### Official Review · Reviewer_xKxN · 2024-10-30

**Soundness:** 2
**Presentation:** 2
**Contribution:** 1
**Rating:** 1
**Confidence:** 5

**Summary:**

In this paper, the authors propose to use a U-Net to further refine the results of the traditional discriminative autoencoders to alleviate the blury output and get crisp and high quality results. Experiments show that the method does improve the visual appearance of the reconstructed images.

**Strengths:**

Adding an additional U-Net and using diffusion model to improve the performance is promising, it should be able to get better results.

**Weaknesses:**

- What's the difference between the proposed method and those utilizing diffusion models for super resolution? I cannot see a big difference.
- Yet the training method is not well explained, do you need to train $E$ and $D_{Initial}$? Or just $D_{refine}$ is trained?
- In Sec. 3.1, it is said that the impact of $D_{Initial}$ is investigated in Table 1, which is not.
- When displaying the reconstruction results, the raw inputs are necessary to see the difference.
- Besides the figures, more quantitative metrics are necessary to justify the method.

**Questions:**

- Overall, the method is trivial and cannot match the high standard of ICLR.
- This is more like an unfinished paper.
- The footnote leaks part of the author information.

---

> ### Author Response · Authors · 2024-11-15
> **Metrics, training details and trivialness**
>
> (1) Difference with super-resolution
> There is a major difference.
> (A) Super-Res: Models take images as inputs and output higher resolution images. This has indeed been done effectively with diffusion models.
> (B) SWYCC: We jointly learn an encoder and a diffusion decoder (Autoencoder) with a continuous latent space. The only commonality is
> that we have “D_inital” that produces an initial  “estimate” of the image. The crucial difference is that all of these components are learnt jointly and there is no frozen stage.
>
> (2) Explaining training:
> Everything is trained jointly, as we note in the caption of Figure 4., should we also elaborate this elsewhere?
>
> (3) Impact of D-Initial in Table 1:
> The only losses that impact d-initial are MSE and Perceptual Loss. When they are set to 0, D-Initial is not adding any information.
> We have elaborated that in the table caption now.
>
> (4) Raw inputs when displaying recon:
> We show raw inputs in Figure 3 (First column is ground truth). Are you suggesting we show them elsewhere ?
>
> (5) More metrics:
> We are going to upload LPIPS, RFID and Inception score shortly.
>
> (6) Method is trivial:
> We strongly disagree with this. Could you elaborate on why this method is trivial? There are a few important things we show in the paper
> that we haven’t seen before
> (a) Making a jointly trained auto encoder with diffusion decoder on continuous latents beat GAN performance.
> (b) Show that such an encoder when paired with latent diffusion can achieve higher compression and better generation performance
> (c) The particular choice of splitting the decoder into d_initial and d_refine with the particular choices of losses (and training them jointly)
> does not exist in literature and we show how that is crucial to beating GANs.
>
> Apart from adding additional evals, could you give us additional feedback that would help make the paper stronger?

---

> ### Author Response · Authors · 2024-11-15
> **Info leakage**
>
> This has been fixed now, thanks for pointing this out

---

> > ### Author Response · Authors · 2024-11-27
> > **More metrics**
> >
> > Additional metrics other reviewers also requested are added to the appendix.

---

### Official Review · Reviewer_4AT2 · 2024-11-01

**Soundness:** 2
**Presentation:** 2
**Contribution:** 2
**Rating:** 3
**Confidence:** 4

**Summary:**

The paper presents a diffusion-based loss for improving the quality of VAE modeling and reconstruction. The authors proposed the new VAE decoder that consists of two main parts, including a Diffusion UNet. The training was conducted with additional diffusion loss. The proposed model was compared with GAN-based loss methods and the authors demonstrate that proposed method yields better results, especially at higher compression rates. Additionally the authors emphasize the importance if the decoder's stochasticity for better details in generated samples.

**Strengths:**

1. The authors provide full explanation of technical detail, including all architecture details and training process.

2. Wide and well-explained ablation studies was conducted to explore the importance of various components of the model. The experiments devoted to exploration of number of denoising steps and cfg sclaes helps to understand the importance of correct choice of these parameters for quality improvements.

**Weaknesses:**

1.There is lack of sufficient metrics for evaluation. It would be better to provide some editional metrics calculation such as LPIPS (https://richzhang.github.io/PerceptualSimilarity/) or Inception Score (https://arxiv.org/abs/1606.03498).
2. Furthermore the authors provided limited evaluation datasets and comparison models. Additional comparisons with some state-of-the-art methods, such as DC-VAE (https://arxiv.org/pdf/2011.10063) or VAEBM (https://arxiv.org/abs/2010.00654) on some other datasets, such as LSUN (https://arxiv.org/abs/1506.03365) or CelebA-HQ-256  , would provide a better understanding of the quality of the model.
3. Only low-resolution data. Conducting further experiments with higher resolution images would be beneficial to understand the capabilities of the model.

**Questions:**

1. It might be important to mention “DiffuseVAE: Efficient, Controllable and High-Fidelity Generation from Low-Dimensional Latents” paper? Could you explain the main difference between this work and your approach? Is it possible to compare your model to DiffuseVae?

2. Is it necessary to train the UNet? Is it possible to use a pre-trained diffusion model without additional training in the refinement part?

3. Nowadays it is important to be able to work with high-resolution images. Can you scale your method to produce high-resolution images?

4. The suggested decoder of the model consists of two main parts, including the additional UNet. Is it possible that your method provides better results because it utilizes a larger model?

---

> ### Author Response · Authors · 2024-11-15
> **Metrics, comparisons, scale concerns**
>
> (1) Metrics:
> We are in the process of adding LPIPS and Inception score to the next revision, thanks for the suggestion
>
> (2) Comparison to DC-VAE
> DC-VAE reconstruction FID at ImageNet 256x256 is 10.57. Our “generative” FID is < 4, Figure 2, this convincingly shows that we are better than this technique. We will include this in a comparison Table.
>
> (3) VAEBM and other datasets
> Since VAEBM did not show ImageNet results, it is hard to compare. We thought ImageNet i=s best suited to demonstrate the capabilities of our model since it closely resembles a “text to image” like setting. We are working  to see if we can get additional datasets evaluated.
>
> (4) DiffuseVAE comparison:
> Thank you for pointing this out, it is indeed a useful addition to our related work. DiffuseVAE train their auto encoder in 2-stages, unlike us who train end-to-end (DiffuseVAE Section 3.2, 2 stage training). Quote “We train Eq. 10 in a sequential two-stage manner, i.e., first optimizing LVAE and then optimizing for LDDPM in the second stage while fixing θ and ψ (i.e. freezing the VAE encoder and the decoder)”
>
> (5) Is it necessary to train Unet:
> No it is not, we could use a pre-trained U-Net; we could consider adding a study about that if its useful.. There is no additional training, all components are trained jointly. We found that training both DInitial and DRefine jointly was important for performance, we have elaborated that now at the end of Section 3.1
>
> (6) High-resolution images:
> All of the components  we use, ConvNets, Unets(https://arxiv.org/abs/2301.11093), transformers are known to scale extremely well with higher resolutions , so we believe our method will scale equally well.
>
> (7) Better results due to larger model?
> Yes, we do get better results due to a larger model, but we think that that is not necessarily a bad thing.  GANs are harder to scale, as we mention in Related work [Kang et al, 2023]. We believe the difficulty of scaling GANs has prevented the field from exploring higher compression autoencoders. We have also elaborated that using more compute is a limitation for our method.

---

> > ### Comment · Reviewer_4AT2 · 2024-11-27
> >
> > Thank you for your response!
> >
> > However, it is still not very clear, why it is important to train end-to-end pipeline. The ablation on comparison end-to-end training, 2-stage training or usage of pretrained diffusion model without additional tuning might be important in this work.

---

> > > ### Author Response · Authors · 2024-11-27
> > > **Additional ablations.**
> > >
> > > These are good suggestions, and part of the reason they are good is because they would be simple to code up.
> > > However, since it is near end of review period now these will have to wait till camera ready.
> > >
> > > But we note that our recipe is still better than the recipe used by previous SoTA autoencoders (GAN based)
> > >
> > > There are 2 outcomes with these ablations
> > > (a) Our recipe is best
> > > (b) One of the tweaks you mentioned gives better results
> > >
> > > In either case, the paper would be a valuable contribution to the field because it shows that you can get higher compression and better generative performance. We hope you consider that while giving your final score.

---

### Official Review · Reviewer_Roop · 2024-11-04

**Soundness:** 3
**Presentation:** 3
**Contribution:** 3
**Rating:** 6
**Confidence:** 5

**Summary:**

This work adds a diffusion loss to the autoencoder training and shows better reconstructoin quality than the prior GAN-based autoencoder. Specifically, the proposed method first has a coarse reconstruction, that is supervised by MSE and LPIPS loss, then a diffusion model refines the coarse reconstruction (jointly trained). The authors show improvement for both reconstruction in different compression rates and generation.

**Strengths:**

**Update after rebuttal:**

```
The authors have addressed my questions. I agree with other reviewers that more visual samples and discussions about related works can be helpful. I will keep my rating as this work shows promising results for the strength of diffusion autoencoders for general images.
```

---


1. Diffusion loss for autoencoder training is an important direction to explore. This work is one of the first works that show promising results.
2. The proposed method outperforms prior GAN-based autoencoder on ImageNet with common metrics CMMD and FID. The trend of compression ratio also shows the advantage of the diffusion loss.
3. The variance visualization is interesting and provides more insights about the what information are sampled.

**Weaknesses:**

1. The 2-stage pipeline makes the propose method a bit less compelling. The autoencoder is still supervised by MSE + LPIPS loss while the diffusion loss is more like for refinement (although it is joint training).
2. The authors claim that the coarse reconstruction is just for speeding-up the training as the diffusion decoder "should converge to true distribution" (L209), I did not find in the paper for neither: (i) experiments that show without LPIPS loss, the model converges to similar performance; (ii) a theory that guarantees autoencoder with only diffusion loss learns similar representation as LPIPS autoencoder. The theory of diffusion models guarantees p(x|z) can be correctly modeled, but LPIPS should have impact on the latent representation z and thus may change the distribution p(x|z). Thus the claim is not well-justified to me.
3. It would be better to have more qualitative visual comparisons. For example, for different compression rates, and with more diverse samples, to justify the improvement of the proposed method. FID may be overly-related to the LPIPS weight.

**Questions:**

What are the number of trainable parameters for the baseline GAN-based method and the proposed method? Is the baseline GAN autoencoder retrained on the same setting? It would be helpful to provide more details about the baseline method.

---

> ### Author Response · Authors · 2024-11-15
> **Necessity of 2-stage, re-wording of speeding up training, Added qualitative comparison**
>
> (1) On 2 stage pipeline:
> We also initially started with 1-stage decoder, but we found that 2 stage decoder was necessary to get good reconstruction FID. We clarified that at the end of Section 3.1 in the new version.
>
> (2) LPIPS and faster convergence.
> This is a good point and we have re-worded that section as “reducing distortion using additional distance metrics”:
>
> (3) Qualitative
> We added 2 more examples to Figure 11 and are collecting more. We show that LPIPS loss makes samples visually better.
>
> (4) Number of parmeters
> Number of parameters, Our decoder has strictly more parameters because D-Initial is like the GAN decoder and D-Refine adds
> On the order of 400M params. We are gathering exact numbers and putting it in a table. We don’t think  this is necessarily a bad thing because we believe we have found a way to use compute to up the level of compression and do better at generation.

---

> > ### Author Response · Authors · 2024-11-27
> > **Parameter counts added**
> >
> > Parameter counts added to latest draft in appendix.

---

### Meta-Review · Area_Chair_coCD · 2024-12-17

**Metareview:**

The paper introduces a diffusion-based loss for training autoencoders, demonstrating improved reconstruction and generative performance over GAN-based methods, particularly at higher compression ratios. Reviewers highlighted concerns regarding the limited scope of evaluations, requesting additional metrics such as LPIPS and rate-distortion-perception tradeoffs, as well as experiments on more datasets and higher-resolution images. Further clarification on the necessity of the two-stage pipeline, computational costs, and broader comparisons with existing baselines is needed to strengthen the paper’s impact and practical relevance.

**Additional Comments On Reviewer Discussion:**

During the rebuttal period, reviewers raised concerns about limited metrics, missing baseline comparisons, dataset scope, and the necessity of the two-stage pipeline. The authors addressed these by adding LPIPS, rFID, and Inception Score, updating Figure 8 with rate-distortion-perception tradeoffs. The main point for my final decision is that in Tab. 8 SWYCC performs worse on PSNR. In order to really see if the proposed approach learns a good latent space we would need to see the performance on a downstream generative modeling tasks.

---

### Decision · Program_Chairs · 2025-01-22

Reject